# Genetic Hypothesis and Pharmacogenetics Side of Renin-Angiotensin-System in COVID-19

**DOI:** 10.3390/genes11091044

**Published:** 2020-09-03

**Authors:** Donato Gemmati, Veronica Tisato

**Affiliations:** 1Department of Morphology, Surgery and Experimental Medicine, University of Ferrara, 44121 Ferrara, Italy; veronica.tisato@unife.it; 2Centre Hemostasis & Thrombosis, University of Ferrara, 44121 Ferrara, Italy; 3University Centre for Studies on Gender Medicine, University of Ferrara, 44121 Ferrara, Italy

**Keywords:** *ACE*1, *ACE*2, RAS-pathway, COVID-19, SARS-CoV-2, prognostic markers, gender-gap

## Abstract

The importance of host genetics and demography in coronavirus disease 2019 (COVID-19) is a crucial aspect of infection, prognosis and associated case fatality rate. Individual genetic landscapes can contribute to understand Severe Acute Respiratory Syndrome Coronavirus 2 (SARS-CoV-2) burden and can give information on how to fight virus spreading and the associated severe acute respiratory distress syndrome (ARDS). The spread and pathogenicity of the virus have become pandemic on specific geographic areas and ethnicities. Interestingly, SARS-CoV-2 firstly emerged in East Asia and next in Europe, where it has caused higher morbidity and mortality. This is a peculiar feature of SARS-CoV-2, different from past global viral infections (i.e., SARS-1 or MERS); it shares with the previous pandemics strong age- and sex-dependent gaps in the disease outcome. The observation that the severest COVID-19 patients are more likely to have a history of hypertension, diabetes and/or cardiovascular disease and receive Renin-Angiotensin-System (RAS) inhibitor treatment raised the hypothesis that RAS-unbalancing may have a crucial role. Accordingly, we recently published a genetic hypothesis on the role of RAS-pathway genes (*ACE1*, rs4646994, rs1799752, rs4340, rs13447447; and *ACE2*, rs2285666, rs1978124, rs714205) and *ABO*-locus (rs495828, rs8176746) in COVID-19 prognosis, suspecting inherited genetic predispositions to be predictive of COVID-19 severity. In addition, recently, Genome-Wide Association Studies (GWAS) found COVID-19-association signals at locus 3p21.31 (rs11385942) comprising the solute carrier *SLC6A20* (Na+ and Cl- coupled transporter family) and at locus 9q34.2 (rs657152) coincident with *ABO*-blood group (rs8176747, rs41302905, rs8176719), and interestingly, both loci are associated to RAS-pathway. Finally, *ACE1* and *ACE2* haplotypes seem to provide plausible explanations for why SARS-CoV-2 have affected more heavily some ethnic groups, namely people with European ancestry, than Asians.

## 1. Introduction

The novel Severe Acute Respiratory Syndrome Coronavirus 2 (SARS-CoV-2), responsible for the coronavirus disease 2019 (COVID-19), enters human cells by binding its spike protein to the membrane receptor angiotensin converting enzyme 2 (ACE2) and interacting with the transmembrane serine protease 2 (TMPRSS2, widely expressed in epithelial cells at the respiratory, gastrointestinal and urogenital levels), leading to unrestrained ACE2 downregulation [1,2]. The main physiological role of ACE2 is to counteract ACE1 actions within the renin-angiotensin-system (RAS), by converting the potent vasoconstrictor angiotensin-II (Ang-II) in the vasodilator Ang1-7, crucial in controlling the local tissue homeostasis by anti-inflammatory, anti-coagulant, anti-proliferative and anti-fibrotic activity [2].

Previous studies in animal models suggested that RAS-antagonists upregulate ACE2 expression [3,4,5,6], though human studies showed conflicting results [7,8,9], raising the concern that higher ACE2 receptor levels might favor SARS-CoV-2 infection. Accordingly, among RAS-antagonists, ACE-inhibitors (ACEi) or angiotensin-receptor blockers (ARBs) have been suspected to be risky in COVID-19 patients [10,11]. Moreover, the observation that patients affected by cardiovascular diseases, hypertension, kidney disease and diabetes, in which RAS-antagonists represent a standard care, are overrepresented among the severest COVID-19 cases, raised concerns on discontinuing or not RAS-antagonist treatment. On the other hand, higher baseline ACE2 levels have been reported in children, young people and women that are characterized with better prognosis, mild symptoms and lower case fatality rate (CFR) during COVID-19 [12,13,14,15]. Of note, ACE2 cellular density or availability is determinant in maintaining normal tissue homeostasis, thus, ACE2 activators or soluble form of recombinant human ACE2 have been suggested to treat COVID-19 patients [16,17,18,19]. Considering that SARS-CoV-2 infection downregulates cellular ACE2 expression, any action aimed to reestablish ACE1/ACE2 balance might contribute in restoring tissue homeostasis and counteracting lung or other organ failure.

To clarify this issue and weight the risk–benefit ratio of RAS-antagonists use in COVID-19, several observational studies and case-control studies have been published reporting no evidence of increased in-hospital death, disease severity or risk of SARS-CoV-2 infection in continuing ACEi or ARBs treatment [20,21,22]. Overall, these observations cannot be considered the proof of concept for promoting such drugs in every COVID-19 patient, though consensus and recommendations coming from several international societies stated not to stop RAS-antagonist treatments in cardiovascular disease patients with COVID-19 [23,24]. In this line, RAS inhibitors in COVID-19 have been recently discussed [2,25].

Interestingly, ACE1 and ACE2 mutual levels, as well as RAS-antagonists’ response, are strongly regulated by common genetic variants in their genes as well as by other genes out from the RAS-pathway. Of note, wide sex and racial differences in the frequency of *ACE1* and *ACE2* gene variants have been reported [26,27], and selected haplotypes cluster in fragile subgroups of patients at high risk of COVID-19 poor prognosis (e.g., male sex, Black ethnicity, cardiovascular disease), and interestingly, they seem to overlap COVID-19 morbidity and mortality rates [2,28,29,30,31,32]. Together with environmental factors, host genetics and ethnic variations could indeed account for the extreme wide range of SARS-Co-2 symptoms observed among different populations. This is the case of South Asia, in which, despite a vast population, restricted health infrastructures and a recent rush in the number of COVID-19 cases, the mortality rate is lower than in Western Europe and North America [27]. Explanations and mechanisms for this assumed “protection” have not been established yet, though *ACE1* and *ACE2* genes have been proposed as a possible underlining mechanistic reason due to their extremely different gene expression and gene haplotypes distribution between Asians and Caucasians [26,27]. However, gene analyses yielded conflicting data and controversial results are still debated considering different ethnicities [2,27,28,32,33]. Realistically, the burden of genetic predispositions should be considered in the contest of the specific environment in which they exert their actions. Accordingly, social variables such as local health systems, government decisions and people’s collective behaviors are expected to have strong part in the global outcome. As recently reported, there is an imperative need for basic research in environmental sciences on SARS-CoV-2 to allow the identification of specific environmental factors modifying infectivity, severity and spread of COVID-19 [34,35]. Aware of the crucial role of these aspects, we will here focus our main attention on the genetic side of such complex circumstance advising for next dedicated specific investigations.

Specifically, we here propose and discuss about a genetic hypothesis on the RAS unbalancing and over-activation after SARS-CoV-2 infection mainly caused by enhanced Ang-II signaling as a one of the primary drivers of COVID-19 pathobiology and severity. These could potentially be the bases for a genetic score of at risk patients, according to their definite genetic background and ethnicity. Although this could explain part of the variance in the spreading of SARS-CoV-2, further dedicated studies aimed at revealing how the different host susceptibility interact with the local environment could help in recognizing the effective dedicated measures to intervene and counteract next waves of virus infection.

## 2. ACE1/ACE2 Balance

The complex dynamic in the formation of angiotensin peptides mainly depends on two antagonist enzymes ACE1 (EC 3.4.15.1), involved in the local vasoconstrictor/proliferative axis (ACE1/Ang-II/AT1 receptor), and ACE2 (EC 3.4.17.23), involved in the vasodilator/anti-proliferative axis (ACE2/Ang1-7/MAS receptor). Most of ACE1 is bound to tissues with lungs (luminal surface of endothelial cells in lungs vasculature), testis and kidney (endothelial cells, mesangial cells and epithelial cells from the proximal tubule and distal regions of the nephron) containing abundant amounts of ACE1, though the circulating enzyme is also present in the blood [36,37]. With regard to ACE2, the protein is mainly localized in lung alveolar epithelial cells and enterocytes of the small intestine as well as in endothelial cells and smooth muscle cells of pulmonary and extra-pulmonary organs [38]. In the context of the RAS pathway, they work in synergy to maintain the hemodynamic stability preserving in turn endothelial and organs integrity. Many other enzymes and receptors are involved in this crucial pathway as summarized in Figure 1. Inherited and/or acquired alterations of the mediators belonging to this pathway might create disequilibrium affecting anti-inflammatory, anticoagulant, anti-proliferative, anti-fibrotic, anti-apoptotic and anti-oxidant activities as during hypertension, cardiovascular and kidney diseases, leading to organ dysfunction and failure. To counteract RAS over-activation and reestablish ACE1/ACE2 balance, ACEi and ARBs show high efficacy in contrasting Ang-II levels/action by increasing Ang 1-7 or blocking Ang-II type 1 receptor (AT1R), respectively (Figure 1).

Several studies assessing the inhibition of Ang-II signaling in pulmonary injury, acute respiratory distress syndrome (ARDS), pulmonary fibrosis and lung damage, confirmed the benefits of ACEi or ARBs treatment by softening cell apoptosis, fibrosis and inflammatory cytokines (CKs) release [39,40,41]. One of the goals of this approach is to restore the physiological ACE1/ACE2 balance in favor of ACE2-derived peptides. During pathological conditions, such as SARS-CoV-2 infection, suppression of ACE2 due to virus binding alters the ACE1/ACE2 balance in favor of Ang-II overproduction, improving ADAM17 activity (a metallopeptidase expressed by most mammalian cells including lung bronchial epithelial cells, vascular smooth muscle cells and macrophages [42]) responsible for ACE2 shedding further decreasing ACE2-membrane in a vicious detrimental loop [2,43]. On the other hand, some authors cautioned against the use of these drugs, hypothesizing that ACEi/ARBs might increase the risk and spreading of infection or develop severe COVID-19 by increasing *ACE2* expression [44]. Other studies pointed on the development of adverse pharmacogenetics interactions and negative side effects such as hypotension, dry cough or hyperkalemia in an appreciable proportion of patients [45].

Two crucial aspects relate to ACEi/ARBs treatments: i) stopping or continuing drugs in already-treated patients regardless of SARS-CoV-2 infection; ii) hypothesizing a repositioning of such drugs in SARS-CoV-2 infected patients to efficiently restore ACE1/ACE2 balance and contrast disease worsening. Accordingly, it has been recommended to not stop RAS-antagonists in already-treated patients [23,24], and recent papers argued that RAS-inhibitors repositioning during COVID-19 emergency does not compare to what has been done for anti-inflammatory, anti-malaria and anti-coagulant drugs [2,45]. Although there is a recognized association between COVID-19 and coagulopathy, and the disease score severity is positively correlated with pro-coagulant markers (e.g., high D-dimer and fibrinogen levels), accurate risk–benefit evaluations should be considered particularly in severe COVID-19 patients. Indiscriminate anticoagulants administration may increase the risk of spontaneous hemorrhage, heparin-induced thrombocytopenia (HITP) and anti-PF4-heparin complex antibodies formation [46,47,48,49,50]. Finally, in addition to the kind of heparin used for patients’ treatment (i.e., LMWH versus HMWH), individual bleeding and thrombotic risk should be considered both during hospitalization and in post-discharge COVID-19 patients [51,52].

Genetically determined ACE1/ACE2 mutual ratios can give back different susceptibilities and predispositions to SARS-CoV-2 progression and infection, responsible in part for different COVID-19 prognosis and mortality, and such information could be useful to pharmacogenetically combine the best drug-user matching and repositioning [2].

## 3. *ACE1* and *ACE2* Genes

*ACE1* gene maps on chromosome 17 (locus 17q23.3), it comprises 26 exons accounting for a full-length gene of about 21 kb, coding for a gene product of 1306 aa (Genbank, NT010783) responsible for the conversion of Ang-I to Ang-II. Several gene variants influence *ACE1* expression accounting for a wide range of levels in different populations. Among the most relevant, a common insertion/deletion (I/D) of 287-bp in the *Alu*-sequence of intron 16, represented by four individual SNPs (rs4646994, rs1799752, rs4340 and rs13447447), modulates *ACE1* expression [53]. The presence of *Alu*-elements may cause an alternative splicing responsible for protein shortening and the loss of one of the two enzyme active sites in the *ACE1* I-allele, while the counterpart D-allele still maintains the two active sites favoring Ang-I to Ang-II formation [54,55]. Accordingly, the D/D genotype shows the highest serum/tissue ACE1 activity, the I/D genotype shows intermediate levels, and the I/I genotype the lowest ACE1 level [56].

A worldwide geographic genetic analysis showed a decline of the D-allele from the highest frequency in African and Arab regions (0.57–0.88), where it appeared as an ancestral allele, towards the lowest frequency in East-Asia (0.12–0.27) with intermediate frequencies in Europe, Australia and America [57]. The D-allele have had some advantages for humans in hot-and-dry environments to retain salt and water. After humans expanded out of Africa, where the climate gradually becomes colder-and-wetter, the propensity to retain salt and water become deleterious, as it increases genetic susceptibility to hypertension and other hypertension-related diseases such as cardiovascular disease and diabetes accordingly with the “thrifty genotype” hypothesis [57]. The Online Mendelian Inheritance in Man (OMIM), a catalogue of human genes and genetic disorders focused on gene–phenotype relationships, reported a study published in 2004 in which the *ACE1* D-allele was considered a genetic predisposition favoring the progression from pneumonia to SARS in Vietnamese patients [56]. Two years before, susceptibility and outcome in ARDS patients were associated with the *ACE* I/D polymorphism [58], although following publications did not confirm this issue [59,60,61]. The recent SARS-CoV-2 pandemic fired-up again the interest on *ACE1* I/D polymorphism, and despite the several recent investigations, definite results are still lacking [2,29,30,31,32], though a recent paper ascribed an *ACE1* I/I-genotype inverse relation to SARS-CoV-2 related mortality, disclosing exciting correspondences between geographic *ACE1* genotype distribution and COVID-19 morbidity and/or mortality [28] (Table 1). Overall, a correlation between *ACE1* I/D genotypes in previous SARS and in SARS-CoV-2 morbidity and mortality exists, ascribing to the D/D genotype the worst prognosis. The European populations have higher *ACE1* D/D genotype and mortality rates due to COVID-19 than Asians. Accordingly, there is an increasing trend from northern Europe to southern Europe of *ACE1* D/D genotype, and geographically moving eastward from Europe to Asia, the *ACE1* D/D genotype rate decreases [57,62]. This coincides with the geographical migration of the modern Homo sapiens out of Africa and with the associated increasing gradient of the *ACE1* I/I genotype. In this scenario, Yamamoto et al. recently reported detailed information on *ACE1* I/I genotype rates among European, Middle Eastern, South Asian and East Asian countries, merging data from the Center for Systems Science and Engineering at Johns Hopkins University detailing *ACE1* I/I genotype frequencies, number of affected cases/population and number of deaths [28].

*ACE2* gene maps on chromosome X (locus Xp22.22) comprise 18 exons, accounting for a full-length gene of approximately 41 kb, coding for a gene product of 805 aa responsible for the conversion of Ang-II to Ang 1-7. One of the *ACE2* isoforms contains an extra exon at the 5′-end (Genbank, NT011757), though no differences in the number of the unique active site of the enzyme there exists. *ACE2* and *ACE1* share 42% homology in the amino acid sequence of the catalytic domain and have similar exon/intron organization, indicating a common ancestor origin. Several gene variants have been listed in the *ACE2* gene [66] also among those amino acids within domains crucial for ACE2 molecule stability and SARS-CoV-2 entry, hypothesizing sex-differences in receptor-virus affinities [2,67]. In addition, no direct evidence supported the existence of genetically resistant ACE2 mutants against SARS-CoV-2 in different populations [68], although a recent analysis on *ACE2* genetic variability highlighted the possibility to link single nucleotide variations to the risk of COVID-19 neurological complications [69]. Rather, single-cell RNA-seq analysis reported that African-American and Whites have a lower *ACE2* expression cell ratio than Asians, and data accounting for gene variant distribution and allele frequency from East Asian populations yielded higher allele frequencies in variants associated with higher *ACE2* expression, suggesting different individual susceptibility to SARS-CoV-2 within different populations [68,70,71].

Among the most relevant polymorphisms influencing ACE2 activity and levels, attention has been paid to the transition G8790A (rs2285666), with the G/G genotype characterized by about 50% expression reduction compared to A/A genotype [72,73]. Interestingly, this SNP is highly represented within Chinese populations (China-MAP, 0.556 and Han-Chinese-South, 0.557), which have the highest allele frequency among those variants within the *ACE2* gene compared to others populations (e.g., Ad Mixed American, 0.336; African, 0.2114; European, 0.235). This corroborates the hypothesis that, as for the *ACE1* gene, selected haplotypes linked to high *ACE2* expression might influence SARS-Cov-2 CFR among different populations according to the protective role of ACE2 against multifactorial thrombosis [74,75], endothelial dysfunction [76] and severe acute lung failure [2,77].

Given the position of G8790A at the beginning of *ACE2* intron 3 (c.439+4G>A), alternative splicing mechanisms affect gene expression [78,79], with two additional intronic SNPs (rs1978124 and rs714205) in strong linkage disequilibrium with rs2285666 [80,81]. It should be taken into account that X-linked genes (i.e., *ACE2*) in presence of detrimental gene variants affecting product activity, as for the one associated with 8790 G-allele, cannot compensate male-carriers with the normal counterpart 8790 A-allele, with males being hemizygotes and females potentially heterozygotes. This is of particular interest in complex diseases with a strong sex-gap in prognosis, as for SARS-CoV-2 infection [2].

*ACE2* G8790A has been explored in the past in association with hypertension in studies mainly performed among China populations observing a higher frequency of the 8790 A-allele in China [81], though several issues still remain inconclusive [82,83]. More recently, five additional SNPs (rs1514283, rs4646155, rs4646176, rs2285666 and rs879922) in the *ACE2* gene have been reported to be associated with essential hypertension in women of the Chinese Han population during the COVID-19 outbreak [84], though the association with hypertension exhibited high heterogeneity and varies with geographical, ethnic and gender also among the Chinese population [85]. In particular, one of these SNPs (rs2285666; C/T) is characterized by a significant different allele frequency among the Italian population with respect to the worldwide population, being strongly underrepresented among Italians (C = 209; T = 35) with respect to both East and South Asians (EAS: C = 354; T = 410 and SAS: C = 374; T = 344) and less if compared to Americans (C = 458; T = 215) or to the rest of the European populations (C = 345; T = 111) [69]. The rs2285666 variant is located in the splice site region of *ACE2* gene, and although the Human Splicing Finder analysis did not predict significant splicing alterations, it was classified as a significant eQTL in several brain tissues [69].

*ACE1* and *ACE2* are key genes in maintaining RAS balance and having significant differences in the worldwide distribution of their genetic variants affecting the respective activities, they should be investigated in combination [2,72,86,87]. A recent report on an *ACE1* I/D and *ACE2* G8790A combined analysis hypothesized that high ACE1 activity (i.e., *ACE1* D/D-genotype) together with low ACE2 activity (i.e., *ACE2* G/G-females or hemizygous G-males) could be considered a high-risk combination for hypertension in presence of cardiovascular risk factors, old age, dyslipidemia and diabetes [72]. The hypothesized increased genetic susceptibility to unbalance ACE1/ACE2 is not a remote possibility because of the high frequency of both variants particularly observed in some populations where SARS-CoV-2 CFR was higher. In that paper, the assumed “at risk combination” (i.e., *ACE1* D-allele plus *ACE2* G-allele) was 13.8% (DD/GG) and 23.0% (DD/GG+GA) among hypertensive females (healthy controls showed 4.9% and 9.9%, respectively) according to a dominant model or a recessive model, respectively, and 56.7% (DD/ID plus G-allele) among hypertensive males. Of interest, the genetic counterpart assumed as “protective combination” (i.e., II/AA homozygous female cases and II/A- hemizygous male cases) was 2.3% and 10%, respectively, accounting for a 10-fold and 5.7-fold difference [72]. Moreover, previous Genome-Wide Association Studies (GWAS) aimed at identifying novel quantitative trait loci regulating ACE1 activity and ACEi pharmacogenetics, recognized new variants in the *ACE1* gene (rs4343) and in the *ABO*-blood group locus (rs495828 and rs8176746) [88], providing an explanation for the correlation of the *ABO*-locus/ACE1-ACE2 mutual levels/RAS-unbalancing/severe COVID-19. This is in line with the recent GWAS that associates locus 3p21.31 (rs11385942), comprising the solute carrier *SLC6A20* (Na+ and Cl- coupled transporter family), and locus 9q34.2 (rs657152) that is coincident with the *ABO*-blood group (rs8176747, rs41302905, rs8176719) to Italian and Spanish severe COVID-19 patients with respiratory failure [89]. Of note, the lead variant (rs11385942) is located in an intergenic region spanning several genes (*SLC6A20*, *LZTFL1*, *CCR9*, *FYCO1*, *CXCR6*, *XCR1*); at least two of them (*LZTFL1*, *CCR9*) are closely located and the risk allele GA of rs11385942 is associated with increased expression of *SLC6A20* and *LZTFL1* in human lung cells. Moreover, authors supported the role of *ABO* in COVID-19 prognosis, considering the presence of neutralizing antibodies against protein-linked N-glycans in the O-group compared to non-O blood groups [89,90] as well as the well-known link existing between the *ABO*-locus and coagulation von Willebrand factor gene expression (i.e., *VWF*; locus 12p13.31), evoking thrombosis and coagulation disturbances as one of the main mechanisms involved. The VWF protein is the carrier of coagulation Factor VIII and high levels of VWF-FVIII complex are a well-known prothrombotic risk factor not only for intrinsic procoagulant properties but also for affecting the natural anticoagulant effects. In terms of biological mechanistic insights, pulmonary endothelial-cells of non-O blood groups are associated with higher VWF protein compared to O-group, accounting for the role that the GWAS ascribed to the *ABO*-locus in COVID-19 [91,92,93].

Though strongly plausible, we suggest to also consider the *ABO* and *SLC6A20* effects on the ACE1/ACE2 mutual levels, on the RAS pathway and on the salt/water homeostasis [2,94,95,96]. The *ABO*-locus also influences the ACEi response [72,94,95,96] and has been found to reduce previous SARS-CoV infection in both pandemic spreading and individual susceptibility, hypothesizing that the O-blood group was at lower risk of infection by natural anti-A and anti-B antibody protection against viruses [97]. Of note, there is a significant sex difference in ACE1 activity/levels, with females showing lower levels compared to males in both healthy and pathological conditions [2,98]. In addition, the *ACE1* I-allele seems overrepresented among females that could be protected [2] and the D-allele expresses higher levels among males, suggesting in them a higher chance of ACE1/ACE2 imbalance during SARS-CoV-2 infection [1,2,99].

Finally, the RAS pathway can also be affected by other genes, including the ACE2-shedder *ADAM17*, which pushes down the ACE2/Ang1-7/Mas axis, *SRY* (Y-chromosome) and *SOX3* (X-chromosome), which upregulate *AGT* and downregulate *ACE2*, *AT2*, *MAS*. Conversely, *SRY* and *SOX3* have an opposite effect on the *REN* promoter [100,101] (Table 2). Altogether, these data ascribe to genes in the RAS pathway a crucial role in COVID-19 prognosis.

## 4. ACEi and ARBs

Rebalancing RAS pathway plays a key role in the management of several pathological conditions including hypertension, cardiovascular disease, heart failure, diabetes and chronic kidney disease. Four groups of classic RAS antagonists act by targeting the Ang-I/Ang-II/AT1-receptor-axis, though new classes of pharmacological compounds targeting newly discovered RAS constituents have been developed [102,103].

The classical drugs interfering with RAS include the ACEi and ARBs directly or indirectly targeting Ang-II, the renin inhibitors targeting both Ang-I and Ang-II and the mineralocorticoid receptor antagonists targeting aldosterone, though many other RAS peptides (Ang 1-7, Ang 1-9, Ang-III, Ang-IV) play significant roles (Figure 1) [102,103].

ACEi activity is based on the competitive inhibition of ACE1 leading to the block of the Ang-I-to-Ang-II transformation and of Bradykinin degradation, a potent peptide with vasodilator activity promoting several effects including prostacyclin and nitric oxide release. Although different ACEi classes show differences in pharmacokinetic and ACE1-binding affinity, the overall effect is ascribable to Ang-II decrease, associated with aldosterone and vasopressin reduction, Ang-I and Bradykinin increase, without effects on Ang-II-receptors or on other RAS peptide actions. Escaping mechanisms associated with prolonged ACEi treatment, aiming at restoring Ang-II levels, have been recognized. The available drugs include Captopril, used in clinical practice since the 1980s, Benazepril, Enalapril, Fosinopril and many others commercially released over time [104].

ARBs have been developed to complete ACEi action and compensate/counteract their limitations and side effects. They block AT1R with high affinity and with a selectivity of about 10000-folds higher for AT1R compared to AT2R. Abrogation of AT1R pathway overall leads to vasodilatation and decrease on peripheral resistance that might be also due to a constant AT2R stimulation, though this relationship is still unclear [105]. ARBs chemical structure differs in the several compounds leading to differences in receptor binding kinetics and in the ability to suppress adrenal aldosterone. The available ARBs include Losartan, developed in the 1990s, Candesartan, reported as the most AT1R selective antagonist, and other compounds such as Eprosartan, Irbesartan, Telmisartan, Valsartan and Olmesartan, showing different clinical and pharmacological features [106]. The concern of continuing or dismissing ACEi and ARBs treatments in COVID-19 patients has become of great relevance since the discovery that SARS-CoV-2 uses ACE2 as main receptor to enter human cells. A recent metanalysis reported that ACEi/ARBs did not worsen COVID-19 severity in hypertensive patients, suggesting a possible underestimation of the protective effects of RAS inhibitors in COVID-19, cautiously leaving open the hypothesis that there might be potential benefits of using ACEi/ARBs in COVID-19 management [107]. Pharmacogenetic investigations in line with personalized and precision medicine approach will shed light on this crucial point, also helping drug repositioning against SARS-CoV-2.

## 5. Conclusions

The observation that severe COVID-19 patients who develop ARDS are more likely to have a history of hypertension, diabetes, cardiovascular disease and RAS inhibitor treatment highlighted the role of RAS genes and raised pharmacogenetics concerns on the safety of such pharmacological approach. However, the impressive amount of literature on this topic published during the first few months of 2020 did not completely account for several potential confounding factors such as comorbidities, concomitant treatments and baseline pulmonary efficiency, as well as sex, age and ethnicity. Sex, age and hormones are of particular relevance as they dramatically affect cardiovascular disease establishment and prognosis, influencing in turn COVID-19 severity [108]. Moreover, individual genetic backgrounds have been scarcely considered. Host genetics undoubtedly has a role in SARS-CoV-2 pathophysiology, influencing both individual susceptibility to infection and disease progression [2,33]. Although no genetically resistant *ACE2* mutant receptors against SARS-CoV-2 have been recognized in different populations, a wide spectrum in symptoms, disease severity and CFR has been described in different countries, and *ACE2* gene has been considered the “first genetic gateway” involved in infection, severity and outcome [109]. Around the world, various countries differently contrasted the SARS-CoV-2 pandemic according to their health care systems. Nevertheless, it has been reported that Europe experienced the highest CFR (9.6%), followed by North America (5.9%) and Asia, this last showing the lowest value (3.5%) despite the pandemic having spread from China [27,110,111]. Interestingly, place- and age-adjusted CFR evaluations in USA, comparing Blacks, Whites, Hispanics and Asians, ascribed to Blacks the highest CFR in crude-, place-, age- and age/place-adjusted analyses, suggesting gene–environment interactions and ethnic disparities, having compared different ethnicities living in the same country [112,113]. Nevertheless, the apparent difference in disease fatalities between countries, central Europe and East Asia, or in the same country among different ethnicities as reported in US, suggests different possible explanations should be taken into account. Among these, demographics, social/cultural behaviors, local hygienic aspects, delay in outbreaks establishment and low virus-testing capacity are variables that may significantly account for the observed gaps in controlling the pandemic, as well as the availability of artificial respiratory and specialized staff [65]. In addition, multiple viral infections and the involvement of associated immuno-virological factors or potential coronavirus resistance gene mutations occurring among East Asians as a result of long-term co-evolution of the virus and host cannot be excluded [65]. Finally, the existence of health inequities among minority populations unmasked how the pandemic greatly affects the most socially and economically disadvantaged people/gender but paradoxically, also countries with universal health coverage are not able to completely protect its inhabitants against high CFR due to COVID-19 pandemic [2,110,112,113,114].

Accordingly, whether or not lung injury will establish after virus infection may depend on several inherited, acquired and social/political concomitant factors. Genetically determined ACE1 and ACE2 levels, by modulating the dynamic of angiotensin peptides, contribute in maintaining RAS balance and organ homeostasis. On the other hand, a pharmacogenetically determined response to RAS inhibitors contributes to overall drug therapeutic efficacy and/or negative side effects establishment.

To further expand our two recently hypothesized genetic mechanisms [2] based on the two recently-suggested pharmacological mechanisms tuning RAS over-activation during virus infection [25], we have here described two possible opposite genetic scenarios. In the first hypothesis, the resulting unbalance may be weakened by protective inherited mechanisms in both *ACE1* and *ACE2* genes, considering “gain of function” gene variants in *ACE2* gene (i.e., rs2285666) and/or “loss of function” gene variants in *ACE1* gene (i.e., rs4646994). This hypothesizes the existence of inherited protective predispositions to counteract ACE1/ACE2 unbalance caused by virus infection. In the second hypothesis, the opposite haplotype condition might indeed exacerbate RAS unbalance due to SARS-CoV-2 infection and be considered a high-risk setting for organ dysfunction and ARDS. A schematic representation of the complementary hypothesized mechanisms is shown in Figure 2. Likewise, over-activated RAS pathway is efficaciously pharmacologically rebalanced by ACEi and ARBs and several pharmacogenetics studies focused on this issue demonstrated how the same drug may lead to different efficacy and safety in different populations, ascribing this to the different geographic distribution of the aforementioned gene variants [72,115,116] recalling the “thrifty genotype hypothesis” [57]. A combination of other particular protective- or at risk-inherited predispositions could also contribute to the unexplained wide range of the clinical manifestations observed among SARS-CoV-2 infected patients (i.e., asymptomatic, paucisymptomatic and severe patients) and help to further improve drug repositioning.

Specific transmission dynamics, in accordance with spatial epidemiology, strongly comply with individual and population genetic landscapes, particularly for complex models of polygenic diseases in which gene-environment interactions have a role [74,75,117,118]. Then, urgent efforts are needed to develop host genetics and genomics networks accounting for ethnic/geographical variations mainly involving RAS, HLA, CKs, TLR, coagulation and complement pathways as recently suggested [109] displaying distinct geographical and population distributions influencing prognosis and susceptibility to several viral infections dramatically useful to prevent or counteract possible SARS-CoV-2 second waves. Our hypotheses in favor of RAS pathway over-activation as one of the main causes in determining COVID-19 susceptibility and severity, is completely in line with the recently GWAS identified associations of *ABO* and *SLC6A20* loci [89] being crucial modifiers of Ang-II local levels and water/salt reabsorption [2,95,119,120,121].

Finally, while waiting for further results from large GWAS and epidemiological studies among different geographic populations, investigations on other target candidate genes affecting crucial pathways should also be performed. Accordingly, extreme-clinical-phenotype-comparisons or extreme-genotype-comparisons within infected SARS-CoV-2 patients will reveal how genetically determined pathways (un)balance may influence COVID-19 progression, treatment response and prognosis useful to early identify individuals at high risk of poor prognosis, improving complication prevention and appropriate drug selection and repositioning.

## Figures and Tables

**Figure 1 genes-11-01044-f001:**
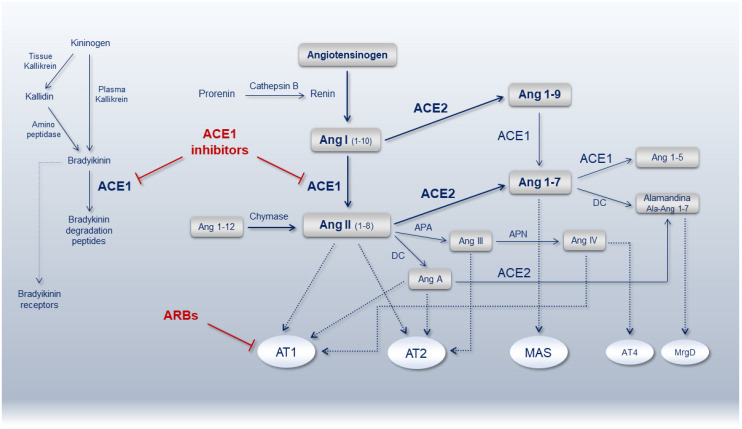
Schematic representation of the renin-angiotensin-system (RAS) pathway showing the main bioactive peptides and key receptors. ACE1, angiotensin-converting enzyme 1; ACE2, angiotensin-converting enzyme 2; Ang, angiotensin; AT1R, Ang-II type 1 receptor; AT2R, Ang-II type 2 receptor; ARBs, angiotensin receptor blockers; MAS, Mas receptor; AT4R, angiotensin receptor type 4; MrgD, MAS-related G protein-coupled receptor member D; APA, aminopeptidase A; APN, aminopeptidase N; DC, decarboxylase.

**Figure 2 genes-11-01044-f002:**
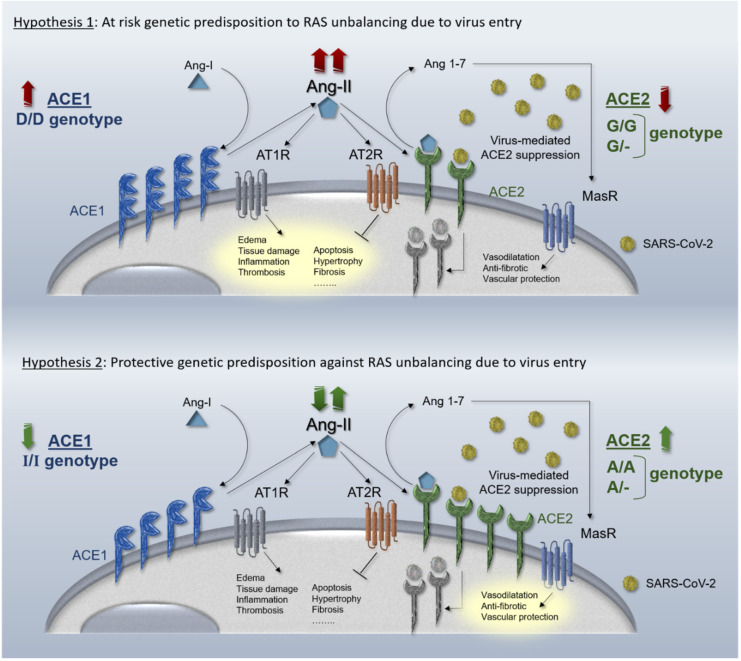
Hypothesized mechanisms of different genetic haplotypes in ACE1 and ACE2 genes in the RAS pathway. Upper panel, the ACE2 downregulation due to virus entry is exacerbated by “loss of function” ACE2 8790 G/G genotype (homozygous G/G females and hemizygous G/- males). The coexistence of ACE1 upregulation due to “gain of function” D/D genotype leads on unrestrained RAS deregulation and ARDS establishment. Lower panel, the ACE2 downregulation due to virus entry is weakened by “gain of function” ACE2 8790 A/A genotype (homozygous A/A females and hemizygous A/- males). The coexistence of ACE1 downregulation due to “loss of function” I/I genotype counteracts RAS unbalancing avoiding ARDS establishment.

**Table 1 genes-11-01044-t001:** Relevant publications dealing with *ACE1* I/D polymorphism in COVID-19.

Type of Manuscript	Geographical Regions (Ethnicity)	Association with SARS-CoV-2 Infection and/or COVID-19 Clinical Phenotype	Ref.
Epidemiological investigation	Asian continent	Positive correlation of D-allele with infection and SARS-CoV-2 mortality rates No significant correlation between D-allele with patient recovery rate	[63]
Ecological Study (meta-analysis)	Worldwide (50 high income countries)	Positive association of D-allele with cumulative incidence and death rate Possible biological plausibility for the association	[64]
Hypothesis paper	Central Europe versus East Asia	ACE1/ACE2 imbalance predicts DD-genotype and D-allele have higher disease prevalence and severity in COVID-19	[65]
Meta-Analysis	Worldwide (30 different countries)	Positive correlation between higher I/D allele frequency ratio and increased disease recovery rate No significant difference in case death rate	[26]
Epidemiological investigation	Europe (25 different countries)	Inverse correlation between D-allele frequency with log-transformed prevalence of COVID-19 infections and mortality	[32]
Epidemiological investigation	European, North-African and Middle East countries (33 countries)	Negative correlation between log-transformed COVID-19 prevalence and associated mortality with D-allele frequency	[31]
Letter to the Editor (Counterpoint)	-	Necessity of considering well-defined geographical spaces to score a polymorphism in a multivariate analysis evaluating its influence on SARS-CoV-2 related infection risk and COVID-19 mortality risk	[29]
Epidemiological investigation	European, Middle Eastern, South Asian and East Asian countries	Negative correlation between II-genotype frequency and SARS-CoV-2 infection rate and number of COVID-19 deaths Increasing trend of II-genotype frequency from European to Asian countries	[28]
Hypothesis paper	-	Sex differences in I/D allele distribution suggest a higher chance of ACE1/ACE2 imbalance among males during ACE2 receptor suppression due to SARS-CoV-2 infection	[2]

**Table 2 genes-11-01044-t002:** Main genes directly or indirectly involved in the RAS pathway and in salt/water homeostasis.

Gene	Name	Locus
*ABO*	α 1-3-N-acetylgalactosaminyltransferase α 1-3-galactosyltransferase	9q34.2
*ACE1*	Angiotensin I converting enzyme 1	17q23.3
*ACE2*	Angiotensin I converting enzyme 2	Xp22.2
*ADAM17*	Metallopeptidase domain 17 TNFα-converting enzyme (TACE)	2p25.1
*AGT*	Angiotensinogen	1q42.2
*AGTR1 (AT1)*	Angiotensin II receptor type 1	3q24
*AGTR2 (AT2)*	Angiotensin II receptor type 2	Xq23
*REN*	Renin	1q32.1
*MAS1*	MAS1 proto-oncogene	6q25.3
*SLC6A20*	Solute Carrier Family 6 Member 20	3p21.31
*SOX3*	SRY-box transcription factor 3	Xq27.1
*SRY*	Sex determining region Y	Yp11.2

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
