# Peer review of "Genetic Hypothesis and Pharmacogenetics Side of Renin-Angiotensin-System in COVID-19"

_genes, 2020, doi:10.3390/genes11091044_

Round 1

Reviewer 1 Report

The authors discuss about a genetic hypothesis on the RAS unbalancing and over-activation after SARS-CoV-2 infection mainly caused by enhanced Ang-II signaling as a primary driver of COVID-19 pathology and severity. The topic is interesting but the following concern needs to be addressed. 

As the authors mentioned in the abstract, "data from previous epidemiological studies share the strong age- and sex-dependent gaps in disease outcomes for COVID-19 as for previous SARS", it seems the novel point of this paper is to deal with these gaps in disease outcomes. However, the authors did not provide insights in it. Therefore, more discussions are needed here or the relevant text should be rephrased in the abstract. 

Author Response

Response to Reviewer 1

Reviewer comment

The authors discuss about a genetic hypothesis on the RAS unbalancing and over-activation after SARS-CoV-2 infection mainly caused by enhanced Ang-II signaling as a primary driver of COVID-19 pathology and severity. The topic is interesting but the following concern needs to be addressed.

As the authors mentioned in the abstract, "data from previous epidemiological studies share the strong age- and sex-dependent gaps in disease outcomes for COVID-19 as for previous SARS", it seems the novel point of this paper is to deal with these gaps in disease outcomes. However, the authors did not provide insights in it. Therefore, more discussions are needed here or the relevant text should be rephrased in the abstract.

Authors reply

According with the Reviewer suggestion, we have modified the Abstract section in order to avoid potential confusion about the main focus of the manuscript. To this end, the sentence mentioned by the Reviewer has been opportunely rephrased (see revised Abstract, lines 20-22).

Reviewer 2 Report

This manuscript “Genetic Hypothesis and Pharmacogenetics Side 3 of Renin-Angiotensin-System in COVID-19: 4 Thrifty Genotype Hypothesis” by Donato Gemmati and Veronica Tisato is hypothesis driven, speculative review. The authors have drawn disparate data from literature to make an argument that ARDS and Covid-19 severity in patients is driven by inherited genetic risk. Here are a few major highlights of this review-

  • The authors have made a strong case of more susceptible ACE1 and ACE11 haplotypes and their prevalence in European population and the resulted disease severity.
  • The severity of lung infection after Covid19 infection in an individual is largely depends on the individuals inherited and acquired concomitant factors. The have provided a good case study of the African-American in USA and why they are severely affected.
  • The authors argued that the use of ACEi and ARBs and elevated disease severity is not true and is dependent on inherited protective ACEi/ACEii ratio.
  • Their rationale to link ACEII with GWAS (published in NEGM) is cogent.

However, there are a few concerns,

1) The authors should  mention about the effects of anticoagulants  causing coagulation abnormalities in severe Covid-19 patients.

2) Because GWAS study link ABO blood group as one of the root causes and since abnormal “coagulation” is a consumptive process, it affects the anticoagulant as well. Therefore, attention should be given to this aspect.

3)  There are some lines throughout the manuscript  that seem redundant and complicated to comprehend; Therefore, the authors should request for help from a professional editor.

Author Response

Response to Reviewer 2

Reviewer comment

This manuscript “Genetic Hypothesis and Pharmacogenetics Side 3 of Renin-Angiotensin-System in COVID-19: 4 Thrifty Genotype Hypothesis” by Donato Gemmati and Veronica Tisato is hypothesis driven, speculative review. The authors have drawn disparate data from literature to make an argument that ARDS and Covid-19 severity in patients is driven by inherited genetic risk. Here are a few major highlights of this review-

  • The authors have made a strong case of more susceptible ACE1 and ACE11 haplotypes and their prevalence in European population and the resulted disease severity.
  • The severity of lung infection after Covid19 infection in an individual is largely depends on the individuals inherited and acquired concomitant factors. The have provided a good case study of the African-American in USA and why they are severely affected.
  • The authors argued that the use of ACEi and ARBs and elevated disease severity is not true and is dependent on inherited protective ACEi/ACEii ratio.
  • Their rationale to link ACEII with GWAS (published in NEGM) is cogent.

Authors reply

We thank the Reviewer for her/his positive comments

However, there are a few concerns,

1) The authors should mention about the effects of anticoagulants causing coagulation abnormalities in severe Covid-19 patients.

Authors reply

Accordingly, the role of anticoagulants causing coagulation abnormalities in severe Covid-19 patients has been included and commented in the revised version of the manuscript (see pages 4-5, lines 144-152, and new references 46-52).

2) Because GWAS study link ABO blood group as one of the root causes and since abnormal “coagulation” is a consumptive process, it affects the anticoagulant as well. Therefore, attention should be given to this aspect.

Authors reply

As suggested by the Reviewer, the link between ABO blood group and coagulation factors (i.e. vWF-FVIII complex) has been mentioned and the effects of the different blood groups on procoagulant and anticoagulant balance has been reported (see page 8; lines 269-278).

3)  There are some lines throughout the manuscript that seem redundant and complicated to comprehend; therefore, the authors should request for help from a professional editor.

Authors reply

The manuscript has been completely revised by an English native speaker and repetition messages have been removed.

Reviewer 3 Report

This is a well written review of the potential role of the RAS pathway in COVID-19, focused to genetic determinants.

Main points.

  • Tables showing specific data of studies evaluating the RAS pathway in COVID-19 patients are encouraged. For example, the studies evaluating the ACE1 I/D polymorphism may not only be referenced but detailed in a table showing number of patients studied, origin, frequency of each genotype, association with clinical phenotype,…. Similarly, a brief description of results concerning the RAS pathway obtained in the WGAS of Ellinghaus et al, must be reported.
  • The description of reference 70 (lines 204-205) must be more specific concerning the role of ACE2 SNPs (particularly rs2285666) in COVID-19.
  • A brief explanation of the association of the ABO blood group (and SLC6A20) with RAS pathway must be supplied.

Minor points.

There are some grammatical mistakes and typos that must be corrected (Line 78: a part; lines 130 recent reviews, ---)

Avoid repetition of messages such as the recommendation of not stopping RAS-antagonists in already treated patients.

Author Response

Response to Reviewer 3

Reviewer comment

This is a well written review of the potential role of the RAS pathway in COVID-19, focused to genetic determinants.

Authors reply

We thank the Reviewer for her/his positive comment

Main points

  • Tables showing specific data of studies evaluating the RAS pathway in COVID-19 patients are encouraged. For example, the studies evaluating the ACE1 I/D polymorphism may not only be referenced but detailed in a table showing number of patients studied, origin, frequency of each genotype, association with clinical phenotype. Similarly, a brief description of results concerning the RAS pathway obtained in the WGAS of Ellinghaus et al, must be reported.

Authors reply

We agree with the Reviewer that a summary-table to collect the studies evaluating the ACE1 I/D polymorphism is useful and it will help the reader. In this line, we added a new Table reporting selected publications dealing with ACE1 I/D polymorphism in COVID-19 patients (see pages 5-6, lines 186-196 and new Table 1 of the revised version of the manuscript).

As requested by the Reviewer, a brief description of results concerning the RAS pathway obtained in the GWAS of Ellinghaus et al, have been included in the revised version of the manuscript (see page 8, lines 262-273 of the revised version of the manuscript).

  • The description of reference 70 (lines 204-205) must be more specific concerning the role of ACE2 SNPs (particularly rs2285666) in COVID-19.

Authors reply

In line with the Reviewer comment, reference 70 (now reference 84) has been deeply commented (see pages 7-8; lines 233-244 of the revised version of the manuscript).

  • A brief explanation of the association of the ABO blood group (and SLC6A20) with RAS pathway must be supplied.

Authors reply

As requested, more insight on the association of the ABO blood group (and SLC6A20) with RAS pathway have been added (see page 8, lines 262-273 of the revised version of the manuscript).

Minor points

There are some grammatical mistakes and typos that must be corrected (Line 78: a part; lines 130 recent reviews). Avoid repetition of messages such as the recommendation of not stopping RAS-antagonists in already treated patients.

Authors reply

The grammatical mistakes highlighted by the Reviewer have been corrected. The manuscript has been completely revised by an English native speaker and repetition messages have been removed.

Reviewer 4 Report

The manuscript entitled "Genetic Hypothesis and Pharmacogenetics Side of Renin-Angiotensin-System in COVID-19: Thrifty Genotype Hypothesis", by Gemmati and Tisato, presents a review on the populational differences in the Renin-Angiotensin-System (RAS), their probable genetic bases and the hypothetisized evolutionary explanation behind (thrifty genotype), and relates this with recent literature on the current SARS-CoV-2 outbreak. The later goes along two main axes: the controversy regarding the effects of ongoing RAS-related drug treatment on the outcome of Covid-19 patients and the few data that has been published on genetic determinants affecting predisposition for severe Covid-19.
The authors try to put together these three lines of research, all of them still having controversies on their own, to come up with a hypothesis of genetic polymorphism on the RAS having a significant impact on the variable response of patients to SARS-CoV-2 infection. This hypothesis was already drafted in a previous article (Gemmati et al, Int. J. Mol. Sci. 2020).
The work of literature review on genetic and populational determinants of COVID-19 outcome is useful, although it may be not that different from that presented on the aforementioned previous article. I think that the proposed hypothesis is interesting, but also feel that the authors have a tendency of over-interpretate the differences in populational COVID-19 outcomes in term of genetic backgrounds, perhaps not acknowledging the differences in health systems, government actions and social behaviour of the population. This differences exist between countries and seem to have a major role on the local pandemics numbers. In my opinion this cannot be left out of the analysis. Expressions of genetic determinism without considering social factor should be avoided, both due to epistemological reasons and because the social variables are expected to be strong covariates in the analysed outcome.
For example, in page 7, line 299, the authors compare CFR between America (clearly dominated by US), Europe and Asia, without any mention to the massive different social and political realities in all these countries. Even more, when talking about the US situation, they interpret differences due to ethnicity, adjusted by place and age, only in terms of genetics. It is known that the access of people to medical care, possibility of complying isolation, etc., are also very important factors to explain that differences.

Minor comments:
1. When authors talk about expression of relevant molecules (ie ACE1, ACE2, TMPRSS2) they should mention in which cell types are these molecules expressed.

2. In page 2, line 62, the authors talk extensively about a retracted article (Mehra et al, N. Engl. J. Med. 2020) and use these results to build an argument. They should not do so, it clearly goes against the spirit of paper retraction. It is not enough to later mention that the article has been retracted.

3. In page 5, line 224, the authors state that one of the two variants (rs11385942) associated with Covid-19 outcome in the Ellinghaus GWAS study (Ellinghaus et al N. Engl. J. Med. 2020) comprises the solute carrier SLC6A20. This association is used to reinforce the RAS-COVID19 connection. However, they should mention that this variant is located in an intergenic regions spanning multiple genes, at least two of them (LZTFL1, CCR9) located as close to the variant as the one they pick.

4. Page 8, line 336:
"Our hypotheses supporting RAS pathway over-activation as one of the main cause in determining COVID-19 susceptibility and severity, is completely in line with..."
The term "supporting" is misleading in this context. Hypothesis do not support facts, but the other way around.

Author Response

Response to Reviewer 4

Reviewer comment

The manuscript entitled "Genetic Hypothesis and Pharmacogenetics Side of Renin-Angiotensin-System in COVID-19: Thrifty Genotype Hypothesis", by Gemmati and Tisato, presents a review on the populational differences in the Renin-Angiotensin-System (RAS), their probable genetic bases and the hypothetisized evolutionary explanation behind (thrifty genotype), and relates this with recent literature on the current SARS-CoV-2 outbreak. The later goes along two main axes: the controversy regarding the effects of ongoing RAS-related drug treatment on the outcome of Covid-19 patients and the few data that has been published on genetic determinants affecting predisposition for severe Covid-19. The authors try to put together these three lines of research, all of them still having controversies on their own, to come up with a hypothesis of genetic polymorphism on the RAS having a significant impact on the variable response of patients to SARS-CoV-2 infection. This hypothesis was already drafted in a previous article (Gemmati et al, Int. J. Mol. Sci. 2020).

The work of literature review on genetic and populational determinants of COVID-19 outcome is useful, although it may be not that different from that presented on the aforementioned previous article. I think that the proposed hypothesis is interesting, but also feel that the authors have a tendency of over-interpretate the differences in populational COVID-19 outcomes in term of genetic backgrounds, perhaps not acknowledging the differences in health systems, government actions and social behaviour of the population. This differences exist between countries and seem to have a major role on the local pandemics numbers. In my opinion this cannot be left out of the analysis. Expressions of genetic determinism without considering social factor should be avoided, both due to epistemological reasons and because the social variables are expected to be strong covariates in the analysed outcome.

For example, in page 7, line 299, the authors compare CFR between America (clearly dominated by US), Europe and Asia, without any mention to the massive different social and political realities in all these countries. Even more, when talking about the US situation, they interpret differences due to ethnicity, adjusted by place and age, only in terms of genetics. It is known that the access of people to medical care, possibility of complying isolation, etc., are also very important factors to explain that differences.

Authors reply

We completely agree to the Reviewer comments and suggestions, and for this reason in the revised version of the manuscript, we firstly recognized the importance of the differences in health systems, government actions and social behavior of the population and countries before and during the COVID-19 pandemic. We acknowledged and linked these crucial points to the main aim of our manuscript focused on the role of possible differences in the genetic landscapes using the main genes ACE1 and ACE2 belonging to the RAS pathway as a paradigm.

In detail, in the Introduction Section (see pages 2-3; lines 82-89) we stated as follow “Realistically, the burden of genetic predispositions should be considered in the contest of the specific environment in which they exert their actions. Accordingly, social variables such as local health systems, government decisions and people collective behaviors are expected to have strong part in the global outcome. As recently reported, there is an imperative need for basic research in environmental sciences on SARS-CoV-2 to allow the identification of specific environmental factors modifying infectivity, severity and spreading of COVID-19 [34,35]. Aware of the crucial role of these aspects, we will here focus our main attention on the genetic side of such complex circumstance advising for dedicated specific investigations”.

In addition, in order not to overestimate the role of genetic, and to contextually ascribe to it a role in viral infection outcome, in the Introduction Section we stated “Although this could explain part of the variance in the spreading of SARS-CoV-2, further dedicated studies aimed at revealing how the different host susceptibility interact with the local environment could help in recognizing the effective dedicated measures to intervene and counteract next waves of virus infection.” (See pages 3; lines 94-97)

In the Conclusion Section, further possible explanations/hypotheses considering gaps in access to medical care have been considered: “Nevertheless, the apparent difference in disease fatalities between countries, central Europe and East Asia, or in the same country among different ethnicities as reported in US, suggests possible explanations to be taken into account. Among these, demographics, social/cultural behaviors, local hygienic aspects, delay in outbreaks establishment and low virus-testing capacity are variables that may significantly account for the observed gaps in controlling the pandemic, as well as the availability of artificial respiratory and specialized staff [65]. In addition, multiple viral infections and the involvement of associated immuno-virological factors or the possibility of coronavirus resistance gene mutations occurring among East Asians as a result of long-term co-evolution of virus and host cannot be excluded [65]. Finally, the existence of health inequities on minority populations, unmasked how the pandemic greatly affects the most socially and economically disadvantaged people/gender and that countries with universal health coverage do not completely protect against COVID-19 pandemic [2,110,112-114]. (See pages 10-11; lines 352-363)

Finally, new dedicated References and appropriate repositioning of already cited references have been included in the revised version of the manuscript (see in particular Refs. 2, 34, 35, 65, 110, 112, 113, 114).

Minor comments

  1. When authors talk about expression of relevant molecules (ie ACE1, ACE2, TMPRSS2) they should mention in which cell types are these molecules expressed.

Authors reply

As suggested by the Reviewer, the main cell types expressing ACE1, ACE2, TMPRSS2 and ADAM-17 have been mentioned in the manuscript (see page 1, lines 39-40; page 3, lines 102-108 and new Refs. 36-38; page 4, lines 130-131 and new Ref. 42).

  1. In page 2, line 62, the authors talk extensively about a retracted article (Mehra et al, N. Engl. J. Med. 2020) and use these results to build an argument. They should not do so, it clearly goes against the spirit of paper retraction. It is not enough to later mention that the article has been retracted.

Authors reply

According to the Reviewer comment on a specific retracted paper (Mehra et al, N. Engl. J. Med. 2020), we firstly deleted the manuscript from the listed references and then completely rephrased the concept in which it was included. As a consequence, concepts and arguments on RAS-antagonists’ usefulness in COVID-19 have been based on the remaining cited references.

  1. In page 5, line 224, the authors state that one of the two variants (rs11385942) associated with Covid-19 outcome in the Ellinghaus GWAS study (Ellinghaus et al N. Engl. J. Med. 2020) comprises the solute carrier SLC6A20. This association is used to reinforce the RAS-COVID19 connection. However, they should mention that this variant is located in an intergenic regions spanning multiple genes, at least two of them (LZTFL1, CCR9) located as close to the variant as the one they pick.

Authors reply

The Reviewer comments and suggestions on the results from the GWAS form Ellinghaus et al N. Engl. J. Med. 2020, on the new locus associated to COVID-19 have been integrated and combined with additional information on the several genes included in the associated region. Accordingly, few comments have been included as follow: “Of note, the lead variant (rs11385942) is located in an intergenic region spanning several genes (SLC6A20, LZTFL1, CCR9, FYCO1, CXCR6, XCR1), at least two of them (LZTFL1, CCR9) are closely located and the risk allele GA of rs11385942 is associated with increased expression of SLC6A20 and LZTFL1 in human lung cells.” (see ACE1 and ACE2 paragraph,  page 8, lines 266-2689)

  1. Page 8, line 336: "Our hypotheses supporting RAS pathway over-activation as one of the main cause in determining COVID-19 susceptibility and severity, is completely in line with..."
    The term "supporting" is misleading in this context. Hypothesis do not support facts, but the other way around.

Authors reply

The term "supporting" has been replaced with “in favour of” (see Conclusion paragraph: page 11; line 395).
